# Soil-transmitted helminth infections among pre-school aged children in Gamo Gofa zone, Southern Ethiopia: Prevalence, intensity and intervention status

Mekuria Asnakew Asfaw[1]*, Tigist Gezmu[1], Teklu Wegayehu[2], Alemayehu Bekele[1], Zeleke Hailemariam[3], Nebiyu Masresha[4], Teshome Gebre[5]

1 Collaborative Research and Training Centre for NTDs, Arba Minch University, Arba Minch, Ethiopia, 2 Department of Biology, College of Natural Sciences, Arba Minch University, Arba Minch, Ethiopia, 3 School of Public Health, College of Medicine and Health Sciences, Arba Minch University, Arba Minch, Ethiopia, 4 Ethiopian Public Health Institute, Addis Ababa, Ethiopia, 5 The Task Force for Global Health, International Trachoma Initiative, Addis Ababa, Ethiopia

* maksambaramr23@gmail.com

**Data Availability Statement:** All relevant data are within the manuscript and its Supporting information files.

## Abstract

### Background

Soil-transmitted helminths (STH), i.e., *Ascaris lumbricoides*, *Trichuris trichiura* and hookworms are among the most prevalent Neglected Tropical Diseases (NTDs) in Ethiopia. Although pre-school aged children pay a high morbidity toll associated with STH infections, evidence on prevalence, intensity and intervention status is lacking in Ethiopia. This study, therefore, aimed to address these gaps to inform decision made on STH.

### Methods

We did a community-based cross-sectional study in five districts of Gamo Gofa zone, Southern Ethiopia; in January 2019. Data were collected using pre-tested questionnaire, and the Kato-Katz technique was used to diagnose parasites eggs in stool. Then, collected data were edited and entered into EpiData 4.4.2, and exported to SPSS software (IBM, version 25) for analysis.

### Results

A total of 2462 PSAC participated in this study. Overall, the prevalence of STH was 23.5% (578/2462) (95% confidence interval (CI) = 21.8%–25.2%). As *caris lumbricoides* was the most prevalent (18.6%), followed by *Trichuris trichiura* (9.2%), and hookworms (3.1%). Of the total, 7.4% PSAC were infected with two STH species. Most of the positive cases with STH showed low infection intensities, while 15.1% *ascariasis* cases showed moderate infection intensities. The study found that 68.7% of PSAC were treated with albendazole. Also, household's level data showed that 39.4% used water from hand-dug well; 52.5% need to travel ≥30 minutes to collect water; 77.5% did not treat water, and 48.9% had no hand

**Funding:** This study is made possible by the generous research grant support of collaborative research and training center for NTDs, Arba Minch University, Ethiopia.

**Competing interests:** The authors have declared that no competing interests exist.

washing facility. In addition, almost 93% care givers achieved less than the mean knowledge and practice score (≤5) on STH prevention.

## Conclusions

This study showed that significant proportions of pre-school aged children are suffering from STH infections despite preventive chemotherapy exist at the study area. Also, gaps in the interventions against STH were highlighted. Thus, a call for action is demanding to eliminate STH among PSAC in Ethiopia by 2030.

## Introduction

Soil-transmitted helminth (STH), i.e., *Ascaris lumbricoides*, *Trichuris trichiura* and hookworms infections are among the most common Neglected Tropical Diseases (NTDs) [1]. It is prevalent mainly in tropical and subtropical areas where water supply, hygiene and sanitation infrastructures are inadequate [2–4]. Moderate and heavy infection intensities of STH are associated with chronic harmful effects on vitamin A and iron status, physical, intellectual, and cognitive development in pre-school aged children (PSAC); these morbidities not only take a huge toll on the health of children, but have also been shown to affect economic development of a nation [5, 6].

Globally, over 2 billion people are affected with STH, where ascariasis accounts for almost 1.2 billion infections, while trichiurasis, and hookworms (*Ancylostoma duodenale* and *Necator americanus*) responsible for over 800 million and 740 million infections, respectively [1, 7, 8]. The global burden of STH infections is estimated at between 5 and 39 million disability-adjusted life years (DALYs), and in 2010, 5.18 million DALYs were estimated as associated with STH infections [9]. The greater burden of STH infections is found in the tropical countries including tropical South America, China, Southeast Asia, and Sub-Saharan Africa [5]. According to the WHO estimate, 42 countries in Africa are endemic for STH with 284 million cases occurring in both school aged and pre-school aged children. These children require periodic administration of preventive chemotherapy [1, 10].

In Ethiopia, the number of people living in STH endemic areas is estimated at 81 million, of which pre-school aged children account for 9.1 million [11]. The greatest numbers of intestinal worms are harbored in children resulting in diarrhea, loss of appetite, weight loss, growth retardation, malnutrition, anemia and cognitive defects [6, 7, 12–14].

In 2020, the World Health Organization (WHO) endorsed a new road map to combat NTDs including STH by 2030, and in fact substantial progress has been made in terms of controlling STH associated morbidity [3]. The WHO goal is to reduce the prevalence of moderate and heavy infection intensities with soil transmitted helminths among preschool and school aged children to below 2%, to make it no longer considered as public health problems by 2030 [3].

In line with WHO's goal, Ethiopia has also set a similar goal to eliminate STH [11]. To achieve these goals in areas where prevalence of any soil-transmitted infection is 20% or higher, periodic mass administration of preventive chemotherapy (deworming) using annual or biannual single-dose albendazole or mebendazole is recommended by WHO for all pre-school and school aged children [15].

As part of the global actions towards Universal Health Coverage (UHC), ending NTDs is prioritized by 2030 in the Sustainable Development Goal (SDG) agenda under target 3.3 [3]. Moreover, working on NTDs helps the vision of universal health coverage, which means that

all individuals and communities access the health services they need without suffering from financial suffering [16].

Since 2008 preventive chemotherapy (PC) against STH in PSAC has been implemented alongside Vitamin A distribution in the study area as well as at national level through community-based drug distribution platform [11]. In 2013, the first national master plan for NTDs was launched, and then the government of Ethiopia has been collaborating with the WHO and other partners for mapping all endemic districts to address SAC through the school-based mass drug administration [17]. Hence, we noted that PC was started before ten years ago to combat STH in PSAC at study area. However, the impact of deworming on STH infection status among PSAC has not been yet monitored and evaluated like SAC, and evidence is lacking at national level, particularly at the study area on prevalence, intensity and intervention status of STH infections among PSAC. Therefore, the present study aimed at to determine prevalence, intensity, and intervention status of STH infections among PSAC in Gamo Gofa zone, and to inform decision-making on STH controls and elimination programs.

## Methods

### Study area and period

This study was conducted in the former Gamo Gofa zone, Southern Ethiopia; in January 2019. The zone is found in Southern Nations, Nationalities, and Peoples' Regional State (SNNPR), and it had 15 districts and two city administrations (CAs). All districts and CAs are endemic for STH, 15 had moderate prevalence and two had low prevalence. According to the 2007 estimate of Central Statistical Agency of Ethiopia, a total of 2,043,668 people live in the zone, of which 1,013,533 are males and 1,030,135 are females [18]. Arba Minch is the capital city of the zone, which is located at 435 km away from Addis Ababa, capital city of Ethiopia.

### Study design and population

A community-based cross-sectional study was conducted. The study population was all selected PSAC in the selected STH endemic *kebeles* (localities).

### Inclusion and exclusion criteria

Since there is no consistent definition of PSAC in the existing literatures, in this study, all children between 1 and 5 years who are not yet attending (primary) school were considered as pre-school children. PSAC who were unable to give stool samples at the time of data collection were excluded from the study. In addition, they were excluded in the event when they were seriously ill or care givers were unable to provide their information.

### Determining sample size and sampling technique

The sample size was determined using single population proportion formula,

$$n = \frac{\left(Z_{\left(\frac{\alpha}{2}\right)}\right)^2 P(1-P)}{d^2};$$

we considered the followings elements to estimate a sample size which could represent the larger population: P, 25.7% (proportion of PSAC infected with STH) [19]; Z, 1.96 at significant level of alpha (α) of 0.05, and desired degree of precision (d) of 3%, and design effect of 3. The computed sample size was 2434, and by adding 10% non-response rate, the total computed sample size was 2678.

Multi-stage cluster sampling technique was employed in order to select study participants (Fig 1). First, districts and CAs STH endemicity status was identified based on findings of previous STH mapping survey conducted at national level (where 2 had low and 15 had moderate prevalence levels) [20]. We excluded 2 districts with low endemicity status since they were not eligible for preventive chemotherapy. Second, 5 districts (Chencha, Dita, Bonke, Deremalo and Demba Gofa) and 12 *kebeles* were selected from 15 districts using simple random sampling technique (SRS). Third, list of eligible households in each *kebeles* which had children between 1 and 5 years of age were identified by health extension workers (HEWs). Finally, one child from each household from each *kebeles* was selected by taking probability proportional-to-population size into account through consecutive home–to-home visit till the required sample size was obtained.

## Study variables and data collection

Variables included in this study were STH infection status (positive or negative for any STH), intensities of infections, socio-demographic and economic characteristics of parent or guardians, social determinants of health, wealth status of households and child related variables and implementation status of STH interventions. Data on these socio-demographic and other variables were collected through face-to-face interviews using pre-tested questionnaire from head of households (HHs) or mothers or guardians. Stool specimens were examined using the WHO recommended Kato-Katz technique under microscopy [21].

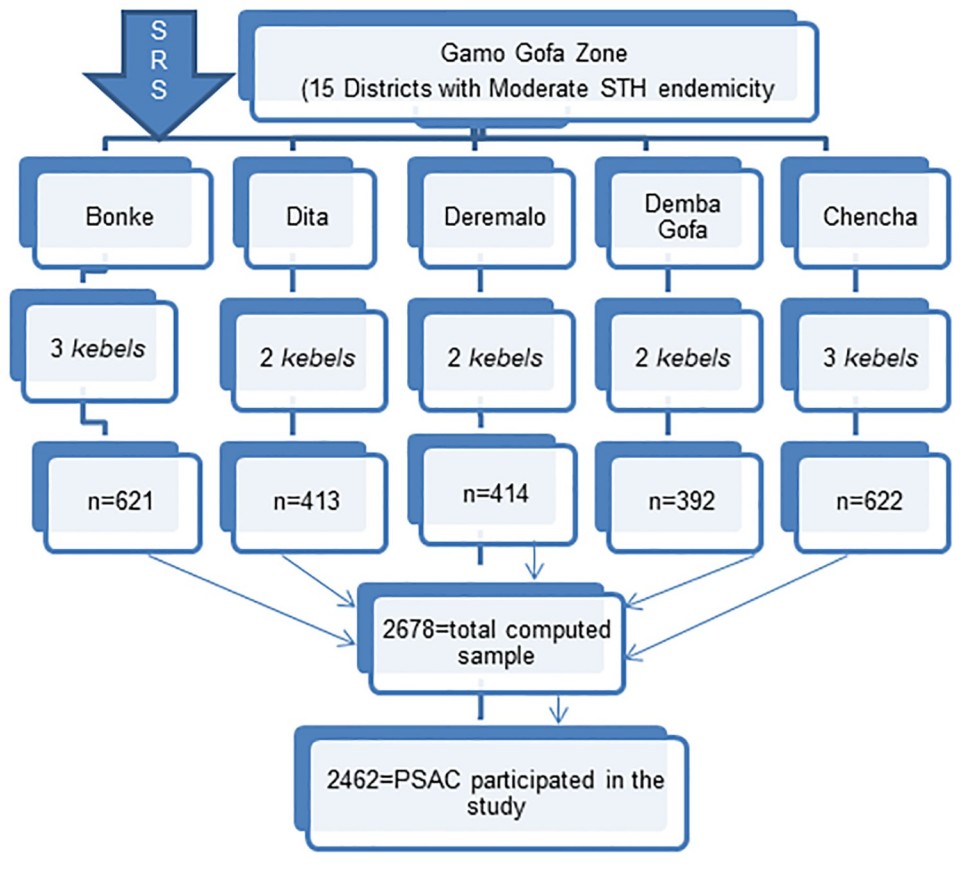

**Fig 1. Sampling profile.**

## Stool collection and processing

Fresh stool specimens were collected using clean, leak proof and screw cup container, and then the collected stool samples were transported in an icebox to nearby health facility for examination. The stool samples were processed within two hours of receipt or saved in an icebox where travel time exceeded two hours. Samples were examined in the local health center by Kato-Katz technique to determine the prevalence and intensities of STH infections.

## Quality control

Data quality was monitored through standard operational procedure, recruiting competent data collectors, pre-testing tools, training data collectors and supervisors, daily checking consistency and accuracy of collected data. The quality of data collection was closely monitored by supervisors. Stools were examined by qualified laboratory technicians. Duplicate slides were prepared per stool sample in order to ensure reliability. For the purpose of bench aid, pictures of parasites eggs were displayed on wall surface in front of microscopy examination for the purpose of internal quality control.

## Statistical analysis and measurements

First, data were edited and entered into EpiData 4.4.2, and then exported to SPSS software (IBM, version 25) for analysis. A difference in prevalence of STH between or among categories of variables was analyzed using Chi-Square test ($X^2$). Summary statistics were computed, and data were presented with tables and graphs.

Wealth analysis was performed as, initially, reliability test was performed using the economic variables involved in measuring the wealth of the households. The variables which were used to compute the alpha value were entered into the principal component analysis. At the end of the principal component analysis, the wealth index was obtained as a continuous scale of relative wealth. Then, quintiles of the wealth index were created. Knowledge and practice on STH transmission and prevention were measured using 11 questions and score was computed by counting value within a case.

In this study, latrine cleanliness was stated as absence of faecal material or any dirt on the upper surface/floor of the latrine, and unsafe water was defined as untreated water obtained from well, river and spring, whereas safe water defined as water obtained from private or public tap water.

We calculated the prevalence by dividing the number of STH positive PSAC by the total number of participants. Intensity of STH infection is the number of helminths (worms) infecting an individual; for each parasite species it was analyzed as light, moderate and heavy infections based on number of eggs per gram of stool (EPG), and it was classified according to the WHO guidelines [22] as follow (Table 1).

According to WHO STH endemicity mapping classifications, there are three categories in line with implementation of mass drug administration (MDA): i) high transmission (where

**Table 1. Criteria for classifying STH infection intensity for each species.**

| STH species | Intensity of infection (EPG) | | |
|---|---|---|---|
| | **Light** | **Moderate** | **Heavy** |
| *A. lumbricoides* | (1–4999) | (5000–49999) | (≥50000) |
| *T. trichiura* | (1–999) | (1000–9999) | (≥10000) |
| Hookworms | (1–1999) | (2000–3999) | (≥4000) |

prevalence is >50%), ii) moderate transmission (where prevalence is between 20%-50%), and iii) low transmission (where prevalence is < 20%) [23, 24].

## Ethics statement

Ethical approval (reference number: CMHS/11222/111) was obtained from Institutional Research Ethics Review Board of Arba Minch University, College of Medicine and Health Sciences, Ethiopia. Oral and written consents were obtained from district administrators and heads of the households before survey was conducted. We obtained consent from parents or guardians as situations dictated. All children that tested positive for one or more STH were promptly treated with albendazole or mebendazole by health workers.

## Results

### Socio-demographic and economic characteristics

Details on socio-demographic and economic characteristics are presented in Table 2. A total of 2462 PSAC participated in this study. Of the total, 246 (10%) were under 2 years of age, and slightly more males participated than females (52% versus 48%). More than half of HHs (57.5%) did not attend any formal education (Table 2).

### Prevalence of STH infections

Of the total surveyed children, 23.5% (578/2462) (95% confidence interval (CI) = 21.8%–25.2%) had at least one type of STH infection. Ascariasis was the most prevalent (18.6%), followed by trichiurasis (9.2%) and hookworms (3.1%). Mixed STH infections (*Ascaris lumbricoides* and *Trichuris trichiura*) found in 7.4% of PSAC. The highest prevalence (33.8%) of any one of STH infection observed in Chencha district, as contrasted to the lowest prevalence (11%) found in Demba Gofa district. In Deremalo district, considerable amount of hookworm infections (10%) were revealed (Fig 2).

The prevalence of STH infections was slightly higher at ≤2 years, while comparing across the age-group (1–5 years). On the other hand, prevalence of any STH infections among females (24.3%) was a little higher than the male's (22.7%), and a higher prevalence of STH infections (24.6%) in rural area was noticed than urban (20.1%) (Table 3).

### STH infections intensity

In majority of STH infections (85%), low infection intensities are associated with hookworms and *Trichuris trichiura* infections, while 15.1% of ascariasis had moderate infection intensity (Table 4). All of the moderate infection intensities were from Chencha and Bonke districts.

### Intervention status against STH infections

**I. Preventive chemotherapy coverage against STH.** The overall self-reported treatment coverage with albendazole (ALB) against STH among PSAC was 68.7% (1691/2462) in the last year before the survey.

**II. Knowledge and practice (KP) of mothers or guardians related to STH transmission and prevention.** Table 5 presents the details on KP on STH transmission and prevention among mothers or guardians. Of the total surveyed mothers or guardians, almost 93% (2283/2462) achieved less than the mean KP score (≤5) on prevention of STH, 42.7% (1052/2462) did not wash hand after defecation, and 77.7% (1913/2462) got information on STH from health extension workers (HEWs) (Table 5).

**Table 2. Socio-demographic data of PSAC and HHs and economic characteristics (N = 2462).**

| Variable | Category | Frequency | Percent (%) |
|---|---|---|---|
| **Child sex** | Male | 1281 | 52 |
| | Female | 1181 | 48 |
| **Child age (years)** | <2 yrs. | 246 | 10 |
| | 3–5 yrs. | 2216 | 90 |
| **Age of HH (years)** | <20 | 22 | 0.9 |
| | 20–29 | 404 | 16.4 |
| | 30–39 | 1284 | 52.1 |
| | 40–49 | 647 | 26.3 |
| | 55–59 | 59 | 2.4 |
| | 60 and above | 46 | 1.9 |
| **Sex of HH** | Male | 2174 | 88.3 |
| | Female | 288 | 11.7 |
| **Educational status** | No formal education | 1416 | 57.5 |
| | Elementary | 508 | 20.6 |
| | Secondary | 279 | 11.3 |
| | Diploma and above | 259 | 10.5 |
| **Occupation** | Farming | 1606 | 65.2 |
| | Employed | 312 | 12.7 |
| | Merchant | 303 | 12.3 |
| | Unemployed | 78 | 3.2 |
| | Daily laborer | 163 | 6.6 |
| **Family size** | <4 | 288 | 11.7 |
| | 4–6 | 1503 | 61.1 |
| | 7 and more | 671 | 27.2 |
| **Residence** | Urban | 628 | 25.5 |
| | Rural | 1834 | 74.5 |
| **Wealth quintile** | Wealthiest | 488 | 19.8 |
| | Wealthy | 498 | 20.2 |
| | Middle income | 482 | 19.6 |
| | Poor | 498 | 20.2 |
| | Poorest | 496 | 20.2 |

**III. Water, sanitation and hygiene (WASH).** Table 6 presents the details on WASH characteristics at HHs level. This study also revealed households data as, 39.4% used water from hand-dug well; 52.5% of need to travel more than 30 minutes to collect water; 77.5% did not use treated water, and 48.9% had no hand washing facility (Table 6).

## Discussion

This study showed operational context specific evidences on prevalence, intensity and intervention status of soil-transmitted helminth infections among PSAC. It is noted that a significant proportion of PSAC are suffering from STH infections despite provision of mass drug administration at the study area. Also, gaps in the intervention status (PC coverage, WASH and KP on STH prevention) against STH were highlighted, which need to be addressed by the STH programs.

In this study, the overall prevalence of STH infections with at least one STH parasite was 23.5%, which would be classified into the moderate transmission category (where prevalence is

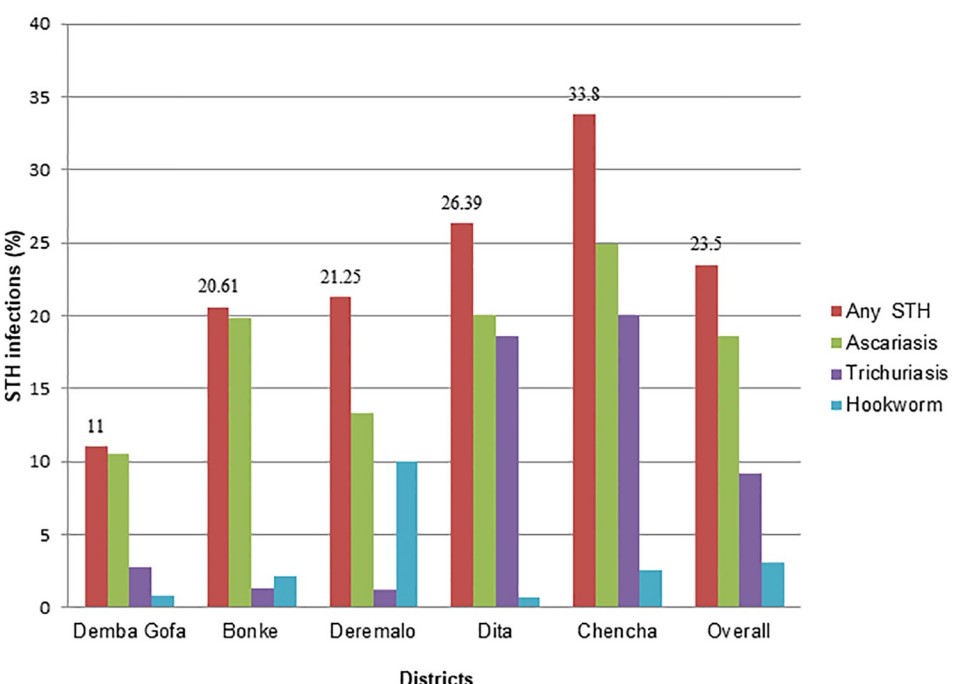

**Fig 2. Prevalence of any STH infection dis-aggregated by districts among PSAC, Gamo Gofa zone, Southern Ethiopia, 2019 (N = 2462).**

between 20% and 50%), and qualifies the requirement of annual STH mass drug administration [24]. Ascariasis was the most prevalent infection (18.6%), followed by trichiurasis (9.2%) and hookworms (3.1%). Most of the positive cases with STH were showed low infection intensities; which could be the positive impact of PC on morbidity reduction. However, 15.1% cases of ascariasis were revealed the moderate infection intensities; This result is much higher than WHO elimination target of STH (which is defined as < 2% proportion of soil-transmitted helminth infections of moderate and heavy intensity due to *A. lumbricoides*, *T. trichuria*, *N. americanus and A. duodenal*) [3]. In our study, despite initiation of preventive chemotherapy about 10 years ago, the prevalence of STH did not show significant reduction probably due to weak implementation of control strategies-social behavioral change communication (SBCC), inadequate mass drug administration coverage and WASH, as highlighted in the findings of this study.

Although the overall prevalence of STH infections observed in this study is comparable with studies conducted in another parts of Ethiopia (Butajira and Wonji) and West China [25–27], our study reached out larger area and powered with sufficient sample size to assess prevalence, intensity and intervention status of STH among PSAC. On the contrary, the STH prevalence in our study is slightly lower than the prevalence reported in Dembiya, northwest Ethiopia [19]. In addition, the prevalence in our study is significantly lower than the findings of other studies conducted in another part of Ethiopia and in Nigeria, Cameroon, Ecuador, Uganda, Kenya and Honduras [28–35]. These differences observed from our study could be due to variation in socio-cultural, social determinants, behavioral characteristics and implementation of prevention and control measures.

In our study, ascariasis was identified as the commonest species of STH, and this finding supports studies done in another part of Ethiopia, Nigeria and China [19, 25, 27, 28]. On the contrary, some other studies conducted in Ethiopia, Ecuador and Honduras showed high

**Table 3. Prevalence of STH infections among PSAC dis-aggregated by different selected variables in Gamo Gofa zone, 2019 (N = 2462).**

| Variable | Category | STH Infection status | | $X^2$-test | P-value |
|---|---|---|---|---|---|
| | | Negative, n (%) | Positive, n (%) | | |
| **Child sex** | Male | 990 (77.3) | 291 (22.7) | 0.354 | 0.366 |
| | Female | 894 (75.7) | 287 (24.3) | | |
| **Child age group (years)** | ≤2 yrs. | 187 (76) | 59 (24) | 0.039 | 0.843 |
| | 3–5 yrs. | 1697 (76.6) | 519 (23.4) | | |
| **Place of residence** | Urban | 502 (79.9) | 126 (20.1) | **5.467** | **0.019** |
| | Rural | 1382 (75.4) | 452 (24.6) | | |
| **Child age (years)** | 1 | 187 (76) | 59 (24) | 0.914 | 0.923 |
| | 2 | 414 (75.1) | 137 (24.9) | | |
| | 3 | 563 (77) | 168 (23) | | |
| | 4 | 522 (77.2) | 154 (22.8) | | |
| | 5 | 198 (76.7) | 60 (23.3) | | |
| **Child soil eating habit** | No | 1518 (76.9) | 457 (23.1) | 0.371 | 0.542 |
| | Yes | 261 (78.4) | 72 (21.6) | | |
| **Family size** | <4 | 220 (76.4) | 68 (23.6) | 2.219 | 0.330 |
| | 4–6 | 1164 (77.4) | 339 (22.6) | | |
| | 7 and above | 500 (74.5) | 171 (25.5) | | |
| **Mothers' (guardians') educational status** | Can't read and write | 774 (76.9) | 233 (23.1) | 5.432 | 0.246 |
| | Can read and write | 291 (73.9) | 103 (26.1) | | |
| | Elementary | 486 (76.1) | 153 (23.9) | | |
| | Secondary | 225 (76.8) | 68 (23.2) | | |
| | Diploma and above | 108 (83.7) | 21 (16.3) | | |
| **Mothers' (guardians') occupation** | Farming | 974 (74.6) | 331 (25.4) | **16.288** | **0.003** |
| | Employed (Gov.) | 166 (81.8) | 37 (18.2) | | |
| | Merchant | 348 (73.7) | 124 (26.3) | | |
| | Unemployed | 334 (82.3) | 72 (17.7) | | |
| | Others* | 62 (81.6) | 14 (18.4) | | |
| **Wealth quintile** | Highest | 379 (77.7) | 109 (22.3) | 1.372 | 0.712 |
| | Fourth | 379 (76.1) | 119 (23.9) | | |
| | Middle income | 374 (77.6) | 108 (22.4) | | |
| | Second | 369 (74.1) | 129 (25.9) | | |
| | Lowest | 383 (77.2) | 113 (22.8) | | |
| **Water source** | Pipe | 891 (79) | 237 (21) | **8.289** | **0.04** |
| | Well | 223 (76.1) | 70 (23.9) | | |
| | Public bono | 721 (74.3) | 250 (25.7) | | |
| | Other** | 49 (70) | 21 (30) | | |

* = Daily laborer and housewife

** = River and spring

**Table 4. STH infection intensity among PSAC in Gamo Gofa zone, Southern Ethiopia, 2019 (N = 2462).**

| Type of STH infection | Mean (EPG) | Infection intensity | | |
|---|---|---|---|---|
| | | Light | Moderate | Total infected PSAC |
| **Ascariasis** | 2152 | 388 (84.90) | 69 (15.1) | 457 |
| **Hookworms** | 154 | 76 (100) | 0 | 76 |
| **Trichiurasis** | 135 | 226 (100) | 0 | 226 |

**Table 5. Knowledge and practice (KP) of mothers or guardians of PSAC related to STH transmission and prevention in Gamo Gofa zone, Southern Ethiopia, 2019 (N = 2462).**

| Variables | Category | Frequency | % |
|---|---|---|---|
| **Knew about STH** | Yes | 2208 | 89.7 |
| | No | 254 | 10.3 |
| **Knew at least one STH transmission (n = 2208)** | Yes | 2153 | 97.5 |
| | No | 55 | 2.5 |
| **Knew at least one STH prevention way (n = 2208)** | Yes | 2135 | 96.7 |
| | No | 73 | 13.5 |
| **KP score on STH transmission** | ≤5 | 2262 | 91.9 |
| | >5 | 200 | 8.1 |
| **KP score on STH prevention** | ≤5 | 2283 | 92.7 |
| | >5 | 179 | 7.3 |
| **Source of information about STH** | Health facility | 271 | 11 |
| | HEWs | 1913 | 77.7 |
| | Radio or TV | 24 | 1 |
| | None | 254 | 10.3 |
| **Habit of washing hand after latrine** | Yes | 1410 | 57.3 |
| | No | 1052 | 42.7 |
| **Habit of washing hand before meal** | Yes | 2347 | 95.3 |
| | No | 2462 | 4.7 |
| **Habit of washing hand after cleaning child** | Yes | 2229 | 90.5 |
| | No | 233 | 9.5 |
| **Habit of washing hand before cooking** | Yes | 2069 | 84 |
| | No | 393 | 16 |
| **Habit of washing fruits or vegetables before eating** | Yes | 1841 | 74.8 |
| | No | 621 | 25.2 |
| **Habit of washing hand after work** | Yes | 2253 | 91.5 |
| | No | 209 | 8.5 |
| **Where do you dispose child's faeces?** | Within home compound | 194 | 7.9 |
| | Toilet | 2002 | 81.3 |
| | Garbage | 266 | 10.8 |

prevalence of Trichiurasis [31, 34, 36], and a study conducted in Uganda showed high prevalence of hookworm [32]. These differences might be related to variation in environmental factors, such as climate, rainfall, topography, surface temperature, altitude, and soil type which have a great impact on the distribution of STH [37]. Moreover, in this study we found significant amounts of mixed infections, 7.4% of PSAC were infected with two STH species (*Ascaris lumbricodes and Trichuris trichiura*); this finding is in line with a study conducted in another areas of Ethiopia and Nigeria [25, 26, 28].

In this study, slightly higher prevalence at age ≤2 years was observed, while comparing across the age-group (1–5 years); the possible explanation related to this difference is due to the fact that current mass drug administration among PSAC often does not include children age ≤2 years. On the contrary, other studies revealed numerical increase in prevalence of STH as age increase [25, 28]. In addition, in this study, prevalence of any STH infection among females (24.3%) was a little higher than the male's prevalence (22.7%), this result supports the findings of a study conducted in another part of Ethiopia [25]. These differences might be due to low access and uptake of preventive chemotherapy among females, as justified by data of our study.

**Table 6. WASH characteristics of households among PSAC participants in Gamo Gofa zone, Southern Ethiopia, 2019.**

| Variables | Category | Frequency | % |
|---|---|---|---|
| Source of water | Pipe | 1128 | 45.8 |
| | Well | 293 | 11.9 |
| | Public bono | 971 | 39.5 |
| | Other* | 70 | 2.8 |
| Distance from water source | < 30 min | 1170 | 47.5 |
| | ≥ 30 min | 1292 | 52.5 |
| Adequate water | No | 536 | 21.8 |
| | Yes | 1926 | 78.2 |
| Habit of treating water | No | 1909 | 77.5 |
| | Yes | 553 | 22.5 |
| Do you have latrine | Yes | 2397 | 97.4 |
| | No | 65 | 2.6 |
| Type of latrine (n = 2397) | Pit | 2360 | 98.5 |
| | Improved pit latrine | 37 | 1.5 |
| Latrine clean | No | 1766 | 73.7 |
| | Yes | 631 | 26.3 |
| Hand washing facility around toilet (functional) | No | 1225 | 51.1 |
| | Yes | 1172 | 48.9 |
| Soap or ash available at hand washing station (n = 1172) | Yes | 298 | 25.4 |
| | No | 874 | 74.6 |
| Reason for absence of latrine (n = 65) | No space | 6 | 9.2 |
| | No money | 10 | 15.4 |
| | No skill | 9 | 13.8 |
| | Did not know importance | 40 | 61.5 |
| If no latrine, where do you defecate (n = 65) | Open field | 56 | 86.2 |
| | Public | 9 | 13.8 |

* = river and spring

In addition, significant proportion (15.1%) of moderate intensity ascariasis was observed in our study, and this finding is higher than results of a study conducted in Butajira, Ethiopia and Honduras [25, 34]. This finding is much higher than WHO elimination target of STH (which is defined as <2% proportion of soil-transmitted helminth infections of moderate and heavy intensities due to *A. lumbricoides*, *T. trichuria*, *N. americanus and A. duodenal*) [3]. The possible explanation related to this difference could be consistency and frequency of mass drug administration that may affect intensity of infections [23].

Furthermore, in our study, gaps in the intervention against STH among PSAC were highlighted. The treatment coverage of ALB in this survey against STH among PSAC was (68.7%); which is lower than the national coverage (71%) of Ethiopia and WHO's target (minimum of 75%) [15, 38]. The possible explanation for the unmet target of PC coverage could be driven by low knowledge of the community regarding the benefits of PC on STH prevention. Most importantly, study participants in this study were pre-school age children, who might not be reached out by the deworming program, especially those ≤ 2 years old. By taking the significant burden of STH among children ≤2 (1–2 years) into consideration, the community based deworming program should reach out these children in collaboration with the community and other stakeholders. On the other hand, obviously, school age children could have better chance to be reached by school-based deworming.

Although site-specific data are required in our case, the importance of WASH interventions to control and eliminate STH reported in different studies [39–41]. However, the status of WASH interventions observed in our study is much lower than WHO's targets that are supposed to be achieved by 2030 [3]. In this study, inadequate WASH infrastructures were observed at household level, where 39.4% were using water from hand-dug well; 52.5% were walking more than 30 minutes to collect water; 77.5% did not treat water, and 48.9% of had no hand washing facility. Of the total surveyed mothers or guardians, almost 93% achieved less than the mean KP score (≤5) on prevention of STH, 42.7% did not wash hand after defecation, and 77.7% got information on STH from health extension workers. The possible reason for these findings could be related to weak and inconsistent social behavioral change communication (SBCC) intervention.

The most outstanding strength of our study is that it is addressing an important national operational research priority with large sample size; which is focusing on parasitological monitoring and control strategies of STH among pre-school aged children.

In this study, there are limitations that need to be taken into account. There might be underestimation of prevalence of STH due to the fact that (1) we collected single stool specimen, which could cause variation in eggs excretion over different times (hours) within a day and across different days; (2) samples were collected from remote villages and there might be rapid desiccation of hookworm eggs in the stool samples, and (3) even though the Kato–Katz technique is sensitive in detecting moderate and high infection intensities (MHI), it has lower detection power, and therefore lower positive predictive values in low-prevalence settings [27, 42].

## Conclusions

Data from our study showed that substantive proportion of pre-school age children in the study area are suffering from STH infections despite provision of preventive chemotherapy distribution at the study area. Also, gaps in the intervention (PC coverage, WASH and KP on STH prevention) to control and eliminate STH were highlighted. Thus, a call for action is demanding to address those gaps, and impact of the interventions should be monitor regularly to achieve the national goal of STH elimination in Ethiopia by 2030. Further, operational research focusing on implementation of PC and impact of specific WASH factors needs to be conducted in different transmission (high, moderate and low) settings to determine more precise epidemiological, environmental and host factors and strengthen STH control and elimination efforts.

## Supporting information

**S1 Questionnaire.**
(DOCX)

## Acknowledgments

Our sincere appreciation and thanks go to study participants, data collectors, supervisors, zonal health office heads, district health office heads, and NTDs focal points in all selected districts of the study area for their kind supports during data collection.

## Author Contributions

**Conceptualization:** Mekuria Asnakew Asfaw.

**Data curation:** Mekuria Asnakew Asfaw, Tigist Gezmu, Teklu Wegayehu, Alemayehu Bekele, Zeleke Hailemariam, Nebiyu Masresha.

**Formal analysis:** Mekuria Asnakew Asfaw, Nebiyu Masresha.

**Funding acquisition:** Mekuria Asnakew Asfaw, Tigist Gezmu, Teklu Wegayehu.

**Investigation:** Mekuria Asnakew Asfaw, Tigist Gezmu, Teklu Wegayehu, Alemayehu Bekele, Zeleke Hailemariam, Nebiyu Masresha, Teshome Gebre.

**Methodology:** Mekuria Asnakew Asfaw, Tigist Gezmu, Teklu Wegayehu, Alemayehu Bekele, Zeleke Hailemariam, Nebiyu Masresha, Teshome Gebre.

**Project administration:** Mekuria Asnakew Asfaw, Tigist Gezmu, Teklu Wegayehu.

**Resources:** Mekuria Asnakew Asfaw, Tigist Gezmu, Alemayehu Bekele, Zeleke Hailemariam, Teshome Gebre.

**Software:** Mekuria Asnakew Asfaw.

**Supervision:** Mekuria Asnakew Asfaw, Tigist Gezmu, Teklu Wegayehu, Alemayehu Bekele, Zeleke Hailemariam, Teshome Gebre.

**Validation:** Mekuria Asnakew Asfaw, Tigist Gezmu, Teklu Wegayehu, Teshome Gebre.

**Visualization:** Mekuria Asnakew Asfaw, Tigist Gezmu, Teklu Wegayehu, Teshome Gebre.

**Writing – original draft:** Mekuria Asnakew Asfaw.

**Writing – review & editing:** Mekuria Asnakew Asfaw, Tigist Gezmu, Teklu Wegayehu, Alemayehu Bekele, Zeleke Hailemariam, Nebiyu Masresha, Teshome Gebre.

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
