## [Decision Letter · Decision Letter 0]

2 Oct 2020

PONE-D-20-25387

Soil-transmitted helminth infections among pre-school aged children in Gamo Gofa zone, Southern Ethiopia: Prevalence, intensity and intervention status

PLOS ONE

Dear Dr. Asfaw,

Thank you for submitting your manuscript to PLOS ONE. After careful consideration, we feel that it has merit but does not fully meet PLOS ONE’s publication criteria as it currently stands. Therefore, we invite you to submit a revised version of the manuscript that addresses the points raised during the review process.

The MS by Asfaw and cols. add information on the situation of STH in preschool children with important description of the consequences of the gaps in the intervention for STH. Despite this study had not a Countrywide coverage it show us the situation of STH is similar in different endemic areas, perhaps the authors could improve it in the discussion. Also, the authors should amend the MS according to the comments made by Reviewer 2.

We look forward to receiving your revised manuscript.

Kind regards,

Marcello Otake Sato, Ph.D., D.V.M.

Academic Editor

PLOS ONE

Journal Requirements:

2. In your Methods section, please provide additional information about the participant recruitment method and the demographic details of your participants. Please ensure you have provided sufficient details to replicate the analyses such as: a) a description of any inclusion/exclusion criteria that were applied to participant recruitment,b) a statement as to whether your sample can be considered representative of a larger population, e) a description of how participants were recruited, and f) descriptions of where participants were recruited and where the research took place.

3. Please provide additional details regarding participant consent. In the ethics statement in the Methods and online submission information, please ensure that you have specified whether consent was informed. If your study included minors, state whether you obtained consent from parents or guardians. If the need for consent was waived, please ensure that you have discussed whether all data were fully anonymized before you accessed them and/or whether the IRB or ethics committee waived the requirement for informed consent.

4. Please include additional information regarding the survey or questionnaire used in the study and ensure that you have provided sufficient details that others could replicate the analyses. For instance, if you developed a questionnaire as part of this study and it is not under a copyright more restrictive than CC-BY, please include a copy, in both the original language and English, as Supporting Information.

Additional Editor Comments (if provided):

The MS by Asfaw and cols. add information on the situation of STH in preschool children with important description of the consequences of the gaps in the intervention for STH. Despite this study had not a Countrywide coverage it show us the situation of STH is similar in different endemic areas, perhaps the authors could improve it in the discussion. Also, the authors should amend the MS according to the comments made by Reviewer 2.

Reviewers' comments:

Reviewer's Responses to Questions

**Comments to the Author**

1. Is the manuscript technically sound, and do the data support the conclusions?

Reviewer #1: Partly

Reviewer #2: Yes

2. Has the statistical analysis been performed appropriately and rigorously? 

Reviewer #1: Yes

Reviewer #2: Yes

3. Have the authors made all data underlying the findings in their manuscript fully available?

Reviewer #1: Yes

Reviewer #2: Yes

4. Is the manuscript presented in an intelligible fashion and written in standard English?

Reviewer #1: Yes

Reviewer #2: Yes

5. Review Comments to the Author

Reviewer #1: The present study aimed to determine prevalence, intensity, and intervention status of STH infections among preschool children (PSAC) in Gamo Gofa zone and to inform decision-making. It is a community-based cross-sectional study in which was included a total of 2462 PSAC. The manuscript is professionally written with well-defined objectives that has been achieved. The methodology is well described with all statistical analyses well done. The conclusions reached by the authors are not different than expected and do not contribute to the improvement of the WHO STH control program, which would be a positive aspect of the work. Although the analyzes are well done, it is a descriptive work with local epidemiological importance It is better placed in a national journal.

Reviewer #2: By and large, the manuscript is written well and achieves the goal of filling in the data gap for STH among the PSAC. There are however typographical errors and grammatical lapses that can be very distracting. The results showed that the high prevalences among the PSAC can come from the gaps in the intervention for STH. The authors should state clearly what these gaps are particularly those that lead to low coverage targets that do not meet recommended targets of the program and of the WHO. The authors could also offer reasons of why such targets are not met maybe using the data that they got from the questionnaires. To strengthen the paper even more, the authors can offer concrete recommendations on how the gaps in the program can be addressed. Their suggestion for what operational research can be done would be very important.

6. PLOS authors have the option to publish the peer review history of their article (what does this mean?). If published, this will include your full peer review and any attached files.

Reviewer #1: No

Reviewer #2: No

---

## [Author Response · Author response to Decision Letter 0]

25 Oct 2020

Authors’ Responses

In all, authors highly valued and appreciate the editor’s and reviewers’ comments. The manuscript has been amended accordingly.

Response to editor’s comments

Journal Requirements:

Response: The manuscript has been amended to satisfy all the journal requirements.

2. In your Methods section, please provide additional information about the participant recruitment method and the demographic details of your participants

Response: We thank the editor for the comments. The manuscript has been modified to address the points raised by the reviewer, lines 134-144.

3. Please ensure you have provided sufficient details to replicate the analyses such as: a) a description of any inclusion/exclusion criteria that were applied to participant recruitment) 

Response: Agreed, the manuscript has been amended to address the points raised by the reviewer, lines 121-125.

4. A statement as to whether your sample can be considered representative of a larger population, e) a description of how participants were recruited, and f) descriptions of where participants were recruited and where the research took place.

Response: Authors are grateful for the comments. The manuscript has been revised to address all points raised by the reviewer, lines 129-130 and 109-116.

5. Please provide additional details regarding participant consent. In the ethics statement in the Methods and online submission information, please ensure that you have specified whether consent was informed. If your study included minors, state whether you obtained consent from parents or guardians. If the need for consent was waived, please ensure that you have discussed whether all data were fully anonymized before you accessed them and/or whether the IRB or ethics committee waived the requirement for informed consent.

Response: We appreciate the editor’s comments, and now correction has been made, lines 195-200.

6. Please include additional information regarding the survey or questionnaire used in the study and ensure that you have provided sufficient details that others could replicate the analyses. For instance, if you developed a questionnaire as part of this study and it is not under a copyright more restrictive than CC-BY, please include a copy, in both the original language and English, as Supporting Information.

Response: The survey questionnaire and laboratory procedure have been included as additional information in the revised manuscript. 

7. The MS by Asfaw and cols. add information on the situation of STH in preschool children with important description of the consequences of the gaps in the intervention for STH. Despite this study had not a Countrywide coverage it show us the situation of STH is similar in different endemic areas, perhaps the authors could improve it in the discussion. Also, the authors should amend the MS according to the comments made by Reviewer 2.

Response: Amendment has been made to improve the manuscript in the discussion section. And now we have made amendment based on the comments made by Reviewer 2.

Response to Reviewers' comments

Reviewer #1

1. is the manuscript technically sound, and do the data support the conclusions?-Partly.

Response: The manuscript has been improved to be technically sound and data support the conclusions.

2. The present study aimed to determine prevalence, intensity, and intervention status of STH infections among preschool children (PSAC) in Gamo Gofa zone and to inform decision-making. It is a community-based cross-sectional study in which was included a total of 2462 PSAC. The manuscript is professionally written with well-defined objectives that have been achieved. The methodology is well described with all statistical analyses well done. The conclusions reached by the authors are not different than expected and do not contribute to the improvement of the WHO STH control program, which would be a positive aspect of the work. Although the analyses are well done, it is a descriptive work with local epidemiological importance It is better placed in a national journal.

Response: Authors are grateful for the feedback and appreciate the reviewer’s view. However, we feel that the findings of this study will be of interest to the wider public health community since STH infections are a global public health problem. Also, studies focusing on STH among PSAC are limited from big African countries like Ethiopia, and we are of the opinion that evidences of our study will certainly add modest value to the existing body of knowledge in STH prevention and control. In addition, we believe that the findings of the study could contribute to improvement of the WHO’s and national STH control programs by providing evidence on STH infection and their intervention status. Moreover, the manuscript has been improved in the discussion section to address some issues raised by the reviewer.

Reviewer #2

1. By and large, the manuscript is written well and achieves the goal of filling in the data gap for STH among the PSAC. There are however typographical errors and grammatical lapses that can be very distracting. The results showed that the high prevalence among the PSAC can come from the gaps in the intervention for STH. The authors should state clearly what these gaps are particularly those that lead to low coverage targets that do not meet recommended targets of the program and of the WHO. The authors could also offer reasons of why such targets are not met maybe using the data that they got from the questionnaires. To strengthen the paper even more, the authors can offer concrete recommendations on how the gaps in the program can be addressed. Their suggestion for what operational research can be done would be very important.

Response: Agreed, all typographical errors and grammatical lapses corrected and amendments have been made in the revised version. In addition, authors highly valued the reviewer’s comments that are raised within the attachment file, and other concerns have been addressed.

---

## [Editor Report · Decision Letter 1]

1 Dec 2020

Soil-transmitted helminth infections among pre-school aged children in Gamo Gofa zone, Southern Ethiopia: Prevalence, intensity and intervention status

PONE-D-20-25387R1

Dear Dr. Asfaw,

We’re pleased to inform you that your manuscript has been judged scientifically suitable for publication and will be formally accepted for publication once it meets all outstanding technical requirements.

Kind regards,

Marcello Otake Sato, Ph.D., D.V.M.

Academic Editor

PLOS ONE

Additional Editor Comments (optional):

The authors have addressed satisfactorily all the comments raised, and now the MS is ready to be accepted.
---

## [Editor Report · Acceptance letter]

4 Dec 2020

PONE-D-20-25387R1 

Soil-transmitted helminth infections among pre-school aged children in Gamo Gofa zone, Southern Ethiopia: Prevalence, intensity and intervention status 

Dear Dr. Asfaw:

I'm pleased to inform you that your manuscript has been deemed suitable for publication in PLOS ONE. Congratulations! Your manuscript is now with our production department. 

Kind regards, 

on behalf of

Dr. Marcello Otake Sato 

Academic Editor

PLOS ONE